# Organophotocatalysed synthesis of 2-piperidinones in one step via [1 + 2 + 3] strategy

Yi-Dan Du[1,4], Shan Wang[1,2,4], Hai-Wu Du[1], Xiao-Yong Chang [1], Xiao-Yi Chen[1], Yu-Long Li[3] & Wei Shu [1,2] ✉

Six-membered *N*-containing heterocycles, such as 2-piperidinone derivatives, with diverse substitution patterns are widespread in natural products, drug molecules and serve as key precursors for piperidines. Thus, the development of stereoselective synthesis of multi-substituted 2-piperidinones are attractive. However, existing methods heavily rely on modification of pre-synthesized backbones which require tedious multi-step procedure and suffer from limited substitution patterns. Herein, an organophotocatalysed [1 + 2 + 3] strategy was developed to enable the one-step access to diverse substituted 2-piperidinones from easily available inorganic ammonium salts, alkenes, and unsaturated carbonyl compounds. This mild protocol exhibits exclusive chemoselectivity over two alkenes, tolerating both terminal and internal alkenes with a wide range of functional groups.

2-Piperidinones are important core substructures in many pharmaceuticals and natural products (Fig. 1a)[1–7] and serve as key precursors or intermediates for the synthesis of multi-substituted piperidines and medicinally relevant compounds[8,9]. In particular, medicinal chemists found that *N*-containing heterocycles, such as 2-piperidinones and derived piperidines are among the second-most prevalent heterocycles in pharmaceutical core structures[1,2,10]. Classical methods to access this structural motif heavily rely manipulation of cyclic precursors, such as the hydrogenation of unsaturated δ-lactams[11–17] and oxidation of piperidines (Fig. 1b)[18,19]. However, the tedious multi-step preinstallation of such backbone structures as well as the strong reductive or oxidative conditions severely hampered the application of these methods. Accordingly, stepwise approaches such as annulations from advanced precursors by reduction-cyclization cascade provided an alternative to access multi-substituted 2-pyridinones[20–22]. Although these strategies are useful, they are limited to specific classes of coupling partners, resulting in specific substituted 2-pyridinones with additional manipulation steps required. To this end, Alper developed a dual catalyzed carbonylation of pyrrolidines by ring expansion to form 6-substituted-2-piperidinones using $[Co_2(CO)_8/Ru_3(CO)_{12}]$[23]. In 2007, Landais reported a $Et_3B/O_2$ mediated multi-component process by involving a tandem radical intermolecular additions-lactamization sequence to access 2-piperidinones[24]. In general, existing methods suffer from limited scope and substitution patterns of 2-piperidinones from advanced synthetic intermediates with poor functional group tolerance. Thus, a streamlined protocol to piperidinones with diverse substitution patterns from easily-available and cheap starting materials is highly desirable yet challenging.

Over the past decades, visible-light enabled chemical bond-forming processes have become an attractive platform for organic synthesis[25–28]. Nicewicz reported the seminal work on alkene activation by photo-initiated single electron oxidation to facilitate hydrofunctionalization with inert nucleophiles (Fig. 1c). However, aliphatic amines are not feasible to undergo such hydrofunctionalization of alkenes due to the low oxidative potentials of starting and resultant amines (Fig. 1c)[29–40]. Recently, our group developed the direct synthesis of aliphatic primary amines enabled by hydroamination of alkenes from ammonium carbonate[41,42]. To date, photocatalytic intermolecular

[1]Shenzhen Grubbs Institute and Department of Chemistry, Southern University of Science and Technology, Shenzhen 518055 Guangdong, P. R. China. [2]State Key Laboratory of Elemento-Organic Chemistry, Nankai University, 300071 Tianjin, P. R. China. [3]College of Chemistry and Environmental Engineering, Sichuan University of Science and Engineering, 643000 Zigong, P. R. China. [4]These authors contributed equally: Yi-Dan Du, Shan Wang. ✉e-mail: shuw@sustech.edu.cn

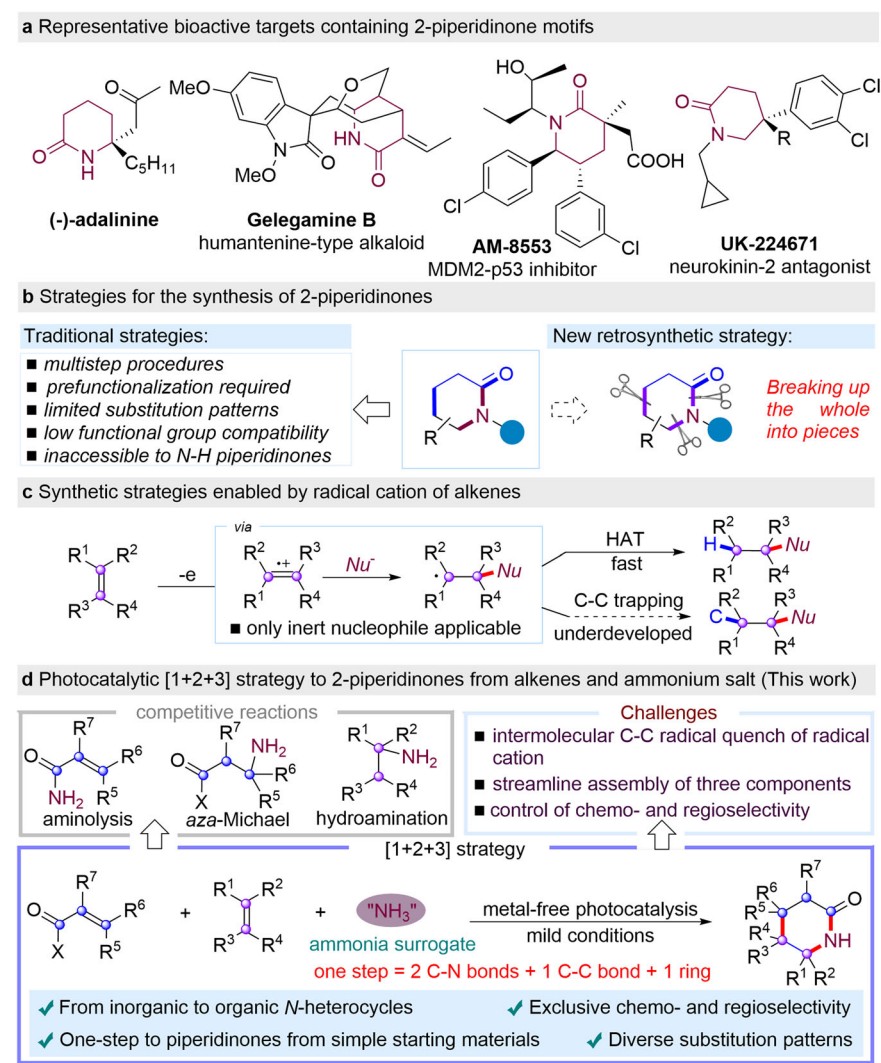

**Fig. 1 | Significance and strategies for the synthesis of 2-piperidinones.**
**a** Representative bioactive targets containing 2-piperidinone motifs. **b** Strategies
for the synthesis of 2-piperidinones. **c** Synthetic strategies enabled by radical
cation of alkenes. **d** Photocatalytic [1 + 2 + 3] strategy to 2-piperidinones from
alkenes and ammonium salt (This work).

functionalizations of alkenes via the cation radical intermediates of
alkenes are limited to hydrofunctionalizations[43], partially due to the
fast hydrogen atom transfer process to quench the stable alkyl radical
species (Fig. 1c). To the best of our knowledge, intermolecular trapping
of such radicals by C-C bond formation remains elusive[44–50]. Thus, we
question the feasibility of quenching the carbon-centered radicals by
intermolecular C−C bond-forming to substantially expand chemical
space of this reaction mode. Herein, we report a metal-free photo-
catalytic [1 + 2 + 3] strategy for the rapid construction of
2-piperidinones from inorganic ammonium salts and alkenes
(Fig. 1d)[51–55]. The use of inorganic salts facilitates the selective C−N
bond formation over two different alkenes as well as the radical trap of
C-C bond-forming process, providing a rapid access to N-unprotected
2-piperidinones from inorganic ammonium salts.

## Results

### Reaction optimization

We commenced our studies with 4-fluoro-β,β-dimethylstyrene **1a** and
methyl α-phenylacrylate **2a** to probe the feasibility of this [1 + 2 + 3]
strategy. After extensive optimization of reaction parameters, the use
of [Mes-3,6-t-Bu₂-Acr-Ph]⁺BF₄⁻ (2.5 mol%) as catalyst, ammonium
acetate (3.0 equiv) as nitrogen source and LiBF₄ (1.0 equiv) as additive
in CH₃CN/PhCl (10:1) under blue LED irradiation at room temperature

was defined as standard conditions (Table 1, entry 1), providing the
desired 2-piperidinone **3a** in 88% yield with a 3.7:1 dr. The structure and
major isomer of **3a** were confirmed by X-ray diffraction analysis. The
use of ammonia surrogate is essential for this reaction. Other ammo-
nium salts could mediate the desired process, albeit leading to the
formation of **3a** in inferior yields (Table 1, entries 2–6 and Supple-
mentary Table 3). Other acridinum-based photocatalyst proved to be
less effective, delivering **3a** in 54–61% yields (Table 1, entries 7–8).
Moreover, additive effect showed a significant impact on the diaster-
eoselectivity. In the absence of LiBF₄, **3a** was obtained in comparable
yield (80% yield) with only 2.6:1 dr (Table 1, entry 9). The use of other
Lewis acids or Lewis bases as additive provided **3a** in lower efficiency
and/or diastereomeric ratios (Table 1, entries 10–14 and Supplemen-
tary Table 8). Control experiments revealed that both photocatalyst
and light irradiation are necessary for the [1 + 2 + 3] transformation
(Table 1, entry 15).

### Substrate scope

After identifying the optimized reaction conditions, we set to explore
the scope of this [1 + 2 + 3] strategy enabled 2-piperidinone synthesis.
First, the scope of radical acceptor was tested and the results are
summarized in Fig. 2. Diverse α-aryl acrylate with *para*- (**3a**–**3e**), *ortho*-
(**3f** and **3g**) and *meta*- (**3h**) substituents on the aromatic rings were all

**Table 1 | Condition evaluation of the reaction[a]**

| Entry | Variation from "standard condition" | Yield of 3a (dr)[b] |
|---|---|---|
| 1 | None | 88% (3.7:1) |
| 2 | $(NH_4)_2CO_3$ instead of $NH_4OAc$ | 41% (1:1.1) |
| 3 | $NH_4HCO_3$ instead of $NH_4OAc$ | 41% (1:1.1) |
| 4 | $NH_2CO_2NH_4$ instead of $NH_4OAc$ | 39% (1:1.2) |
| 5 | $NH_4Cl$ instead of $NH_4OAc$ | trace |
| 6 | $NH_4BF_4$ instead of $NH_4OAc$ | trace |
| 7 | Catalyst **A** instead of Catalyst **C** | 61% (2.1:1) |
| 8 | Catalyst **B** instead of Catalyst **C** | 54% (3.6:1) |
| 9 | w/o $LiBF_4$ | 80% (2.6:1) |
| 10[c] | $Sc(OTf)_3$ instead of $LiBF_4$ | 56% (3.3:1) |
| 11[c] | $Zn(OTf)_2$ instead of $LiBF_4$ | 73% (3.7:1) |
| 12 | 2,6-$tBu_2$pyridine instead of $LiBF_4$ | 88% (2.7:1) |
| 13 | 2,6-lutidine instead of $LiBF_4$ | 82% (2.5:1) |
| 14 | collidine instead of $LiBF_4$ | 69% (2.9:1) |
| 15 | w/o PC, w/o light | N.D. |

[a]The reaction was conducted using **1a** (0.1 mmol), **2a** (0.2 mmol) under indicated conditions.
[b]Yield and diastereomeric ratio (dr) were determined by [1]H NMR of crude mixture of the reaction using PhTMS as internal standard.
[c]Additive (20 mol%) was used.

well-tolerated, giving the desired 2-piperidinone products in excellent yields (71–99%). In addition, acrylamides could be employed as the acceptor to undergo coupling and cyclization to give **3a** in 60% yield. Moreover, fused aryl, alkyl, fluoro-, and benzyl substituted acrylates were all good substrates, affording diverse substitution patterns at 3-position of 2-piperidinones (**3i**–**3m**) in good yields. Notably, esters, terminal and internal alkenes, terminal and internal alkynes were tolerated (**3n**–**3r**) under this metal-free conditions. The reaction underwent chemoselective [1 + 2 + 3] reaction to deliver desired products (**3n**–**3r**) in 49–97% yields, leaving chemical space for further elaboration. Halides and ether containing acrylates were successfully involved in the reaction, furnishing corresponding 2-piperidinones (**3s**–**3v**) in 56–96% yields. Amides without or with free protons are both good substrates in the reaction, delivering **3w** and **3x** in 69% and 91% yields, respectively. Interestingly, vinyl γ-lactone like Tulipalin A could be applied to this reaction, giving **3y** in 79% yield. The structure of **3y** was confirmed by X-ray diffraction analysis. Methylene β-lactam underwent the desired reaction to furnish **3z** in 84% yield. Non-polarized electron-deficient alkenes were also compatible under the reaction conditions, delivering corresponding N-phenyl-2-(4,4,5,5-tetramethyl-2-oxopyrrolidin-3-yl)acetamide **3aa** in 99% yield. Next, the scope with respect to the other alkenes was examined under the standard conditions. A wide range of functional groups and diverse substitution patterns were amenable in the reaction (Fig. 3). Various 2,2-dimethyl-styrenes with electron-donating or electron-withdrawing groups reacted smoothly with different alkene acceptors to furnish desired [1 + 2 + 3] products (**4a**–**4n**) in exclusive regioselectivity and good yields with 2.2:1-6.6:1 dr. Notably, both configurations of stilbene were applicable in the reaction, delivering 3,5,6-trisubstituted 2-piperdinone **4o** in 72% and 55% yields with identical diastereomeric ratio.

1-Substituted styrenes with diverse substitution patterns on arenes are good substrates in the reaction. Electron-donating and electron-withdrawing groups at *para*-, *meta*- and *ortho*-position of arenes are all tolerated, affording 3,5-disubstituted 2-piperidinones (**5a**–**5n**) in synthetic useful yields. 1,1-Disubstituted styrenes could be applied to the reaction to furnish 3,5,5-trisubstituted 2-piperidiones (**5o**–**5s**) in moderate yields. Moreover, the reaction underwent chemoselective functionalization between multiple alkenes (**5q**–**5s**). Furthermore, free alcohol could also be tolerated in the reaction (**5t**). Notably, vinyl silyl ether was successfully involved in the reaction to give bicyclic 2-piperidinone **6a** in 41% yield. Cyclic and acyclic vinyl ethers are both reactive in the reaction, affording bicyclic piperidinone **6b** in 73% yield as single diastereomer and 3,5-disubstituted 2-piperidinone **6c** in 62% yield. It deserves mentioning that aliphatic alkenes were applicable in the reaction. Cyclic aliphatic alkene was converted to octahydro-2H-cyclopenta[b]pyridin-2-one (**6d**) in 53% yield. Acyclic alkyl alkenes reacted to give 3,5,5,6-tetrasubstituted 2- piperidinones (**6e** and **6f**) in 56% and 52% yields. Furthermore, the [1 + 2 + 3] strategy was applied to late-stage functionalization of complex molecules. Alkenes derived from natural products, such as estrone, menthol, and α-cedrene were all compatible with the reaction conditions, successfully affording corresponding natural product-based 2-piperidinones (**7a**–**7c**) in 48%-62% yields. Moreover, the reaction could be scaled up to 2.0 mmol, affording **7c** in 38% yield. The major isomers of compounds (**4j**, **4o**, **6a**, **6b**, **6e**, and **7c**) were confirmed by X-ray diffraction analysis.

## Control experiments and mechanistic consideration

Next, a series of control experiments were carried out to shed light on the reaction mechanism (Fig. 4). First, the reaction of 4-(2-methylprop-1-en-1-yl)-1,1′-biphenyl with methyl 2-(4-methoxyphenyl)acrylate was conducted in the presence of a radical scavenger TEMPO under otherwise identical to standard conditions (Fig. 4a, see more information in Supplementary Information). The

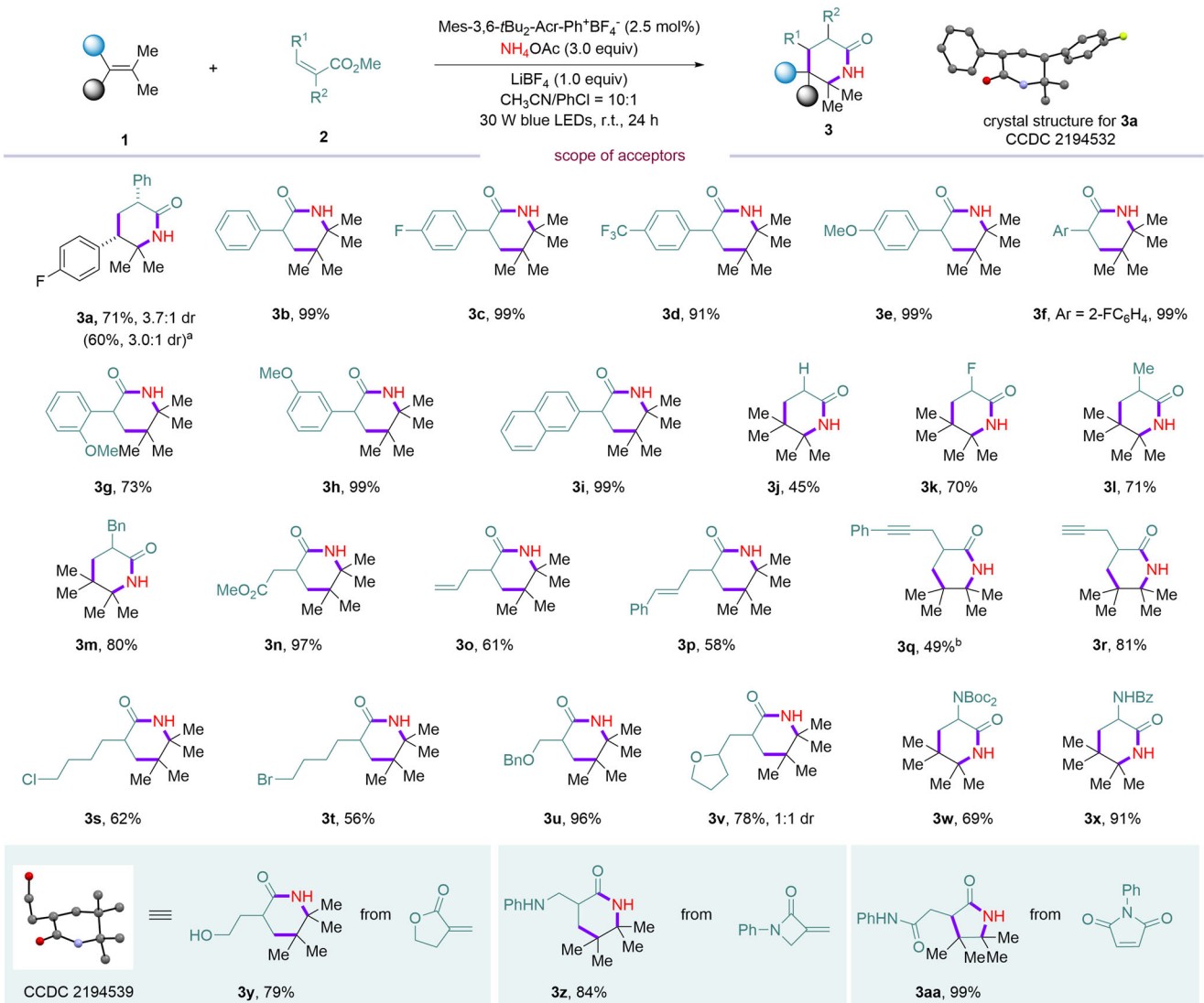

**Fig. 2 | Scope of the acceptors.** The reaction was conducted using **1** (0.1 mmol), **2** (0.2 mmol) under standard conditions unless otherwise noted. Diastereomeric ratio (dr) was determined by $^1$H NMR of crude mixture of the reaction. Isolated yield after flash chromatography. The disordered part, solvent molecules and hydrogen atoms of crystal structures have been omitted for clarity, for details, see Supplementary Figs. 15 and 16. [a]Corresponding Weinreb amide was used. [b]5 mol% photocatalyst was used.

desired [1 + 2 + 3] reaction was completely shut down, suggesting the involvement of radical nature in the reaction process. The TEMPO-trapped adduct **8** could be observed by HR-MS analysis. Second, an experiment between **1b** and methacrylamide was conducted under standard conditions (Fig. 4b). However, no desired product **3l** was detected, excluding the possibility of preformation acrylamides during the reaction course. Furthermore, the light on-off experiments of **1b** and methyl methacrylate was conducted under standard conditions (Fig. 4c). The results indicated the reaction undergo a catalytic process instead of a radical chain pathway. In addition, fluorescence-quenching experiments were conducted to further probe the reaction mechanism using 4-fluoro-β,β-dimethylstyrene (**1a**) and methyl 2-(4-fluorophenyl)acrylate (**2b**) (Fig. 4d). Stern-Volmer analysis and time-resolved fluorescence spectroscopy ($K_{sv}$ = 48.07 $M^{-1}$, $k_q$ = 4.34 × 10$^9$ $M^{-1}$ s$^{-1}$, see Supplementary Information) indicated that this reaction may proceed through a reductive quenching mechanism of Mes-3,6-$t$Bu$_2$-Acr-Ph$^+$BF$_4^-$ by 4-fluoro-β,β-dimethylstyrene. The quantum yield (Φ) of the reaction using **1b** and **2b** was determined to be 0.75 (Fig. 4e), indicating the reaction may undergo a catalytic radical process. Yet, a slow chain propagation mechanism cannot be ruled out at this stage [56].

Based on the experimental results and literature precedence[29-44], a plausible reaction mechanism was proposed and depicted in Fig. 5. First, excited **PC*** was generated from **PC** by visible light irradiation. **PC*** interacted with the alkenes via single electron oxidation to give radical cation intermediate **M1** in conjunction with reduced photocatalyst species **PC-1**. **M1** is trapped by ammonia released from NH$_4$OAc to deliver intermediate **M2**. **M2** could undergo radical addition with acrylates to generate **M3** by C–C bond-formation, which could be further reduced by **PC-1** to give intermediate **M4** and regenerate **PC**. Finally, the intramolecular lactamization of **M4** generated the desired 2-piperidinone products.

## Discussion

In conclusion, an organophotocatalysed [1 + 2 + 3] strategy for the modular access to unprotected 2-piperidinones has been developed at room temperature. The use of an inorganic ammonium salt as ammonia surrogate enables the construction of *N*-containing heterocycles from two different alkenes with exclusive chemoselectivity. The reaction forges two C–N bonds and one C–C bond sequentially in one step to construct 2-piperidinones with diverse substitution patterns, providing a streamlined access to

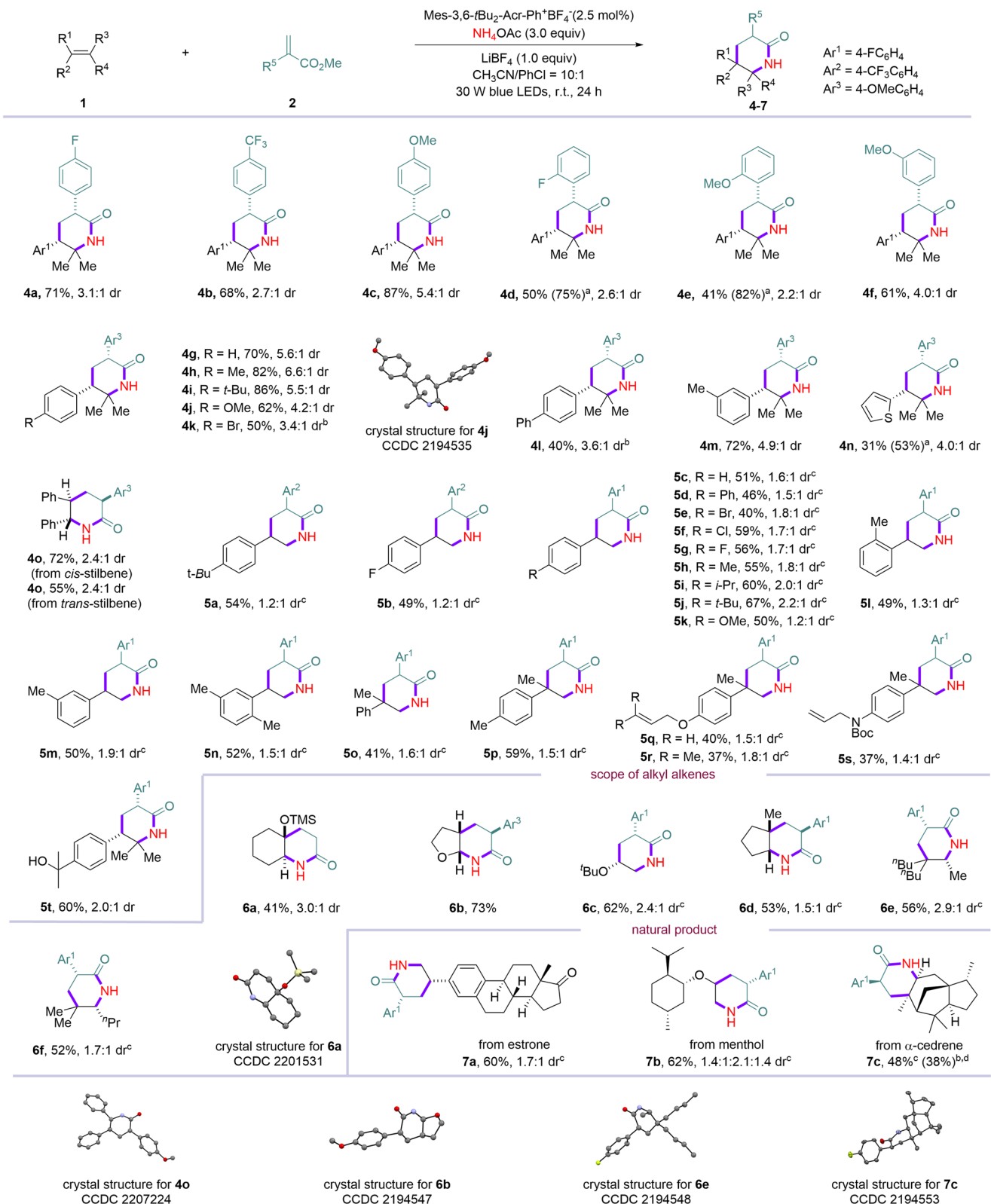

**Fig. 3 | Scope of alkenes.** The reaction was conducted using **1** (0.1 mmol), **2** (0.2 mmol) under standard conditions unless otherwise noted. Diastereoisomeric ratio (dr) was determined by [1]H NMR of crude mixture of the reaction. Isolated yield after flash chromatography. The disordered part, solvent molecules and hydrogen atoms of crystal structures have been omitted for clarity. For more details, see

Supplementary Figs. 17–24. [a] Yield based on the recovery of alkenes. [b] 5 mol% PC was used. [c] The reaction was conducted using **1** (0.1 mmol), **2** (0.2 mmol), PC (5 mol %), NH$_4$OAc (0.3 mmol), LiBF$_4$ (0.1 mmol) in CH$_3$CN:PhCl = 100:1 (10.0 mL CH$_3$CN). [d] The reaction was conducted on 2.0 mmol scale.

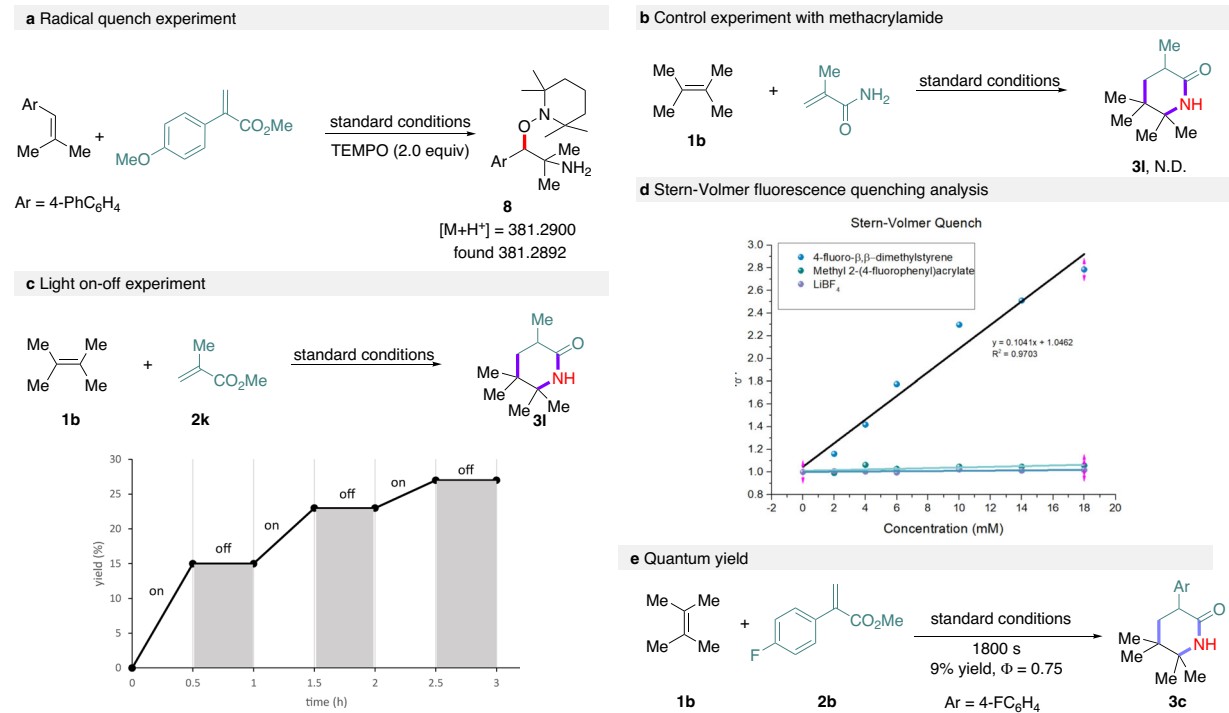

**Fig. 4 | Control experiments and mechanistic investigations. a** Radical quench experiment. **b** Control experiment with methacrylamide. **c** Light-on-off experiment. **d** Stern-Volmer fluorescence-quenching analysis. **e** Quantum yield.

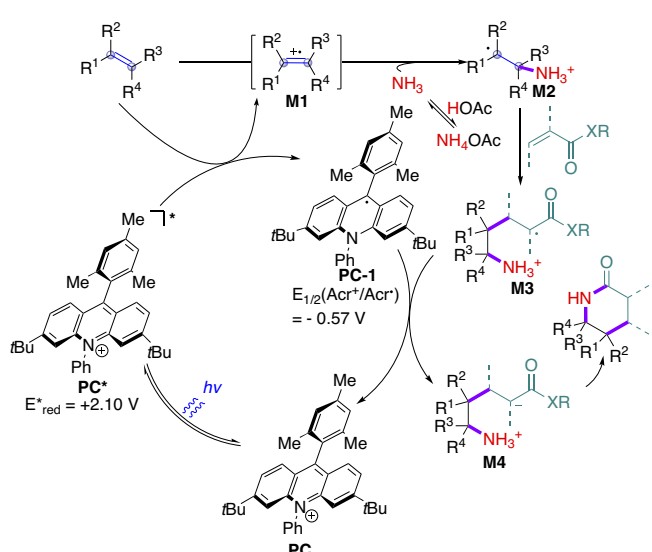

**Fig. 5 | Proposed mechanism.** The plausible mechanism for photocatalytic synthesis of 2-piperidinones from alkenes and ammonium salt by [1 + 2 + 3] strategy.

2-piperidinones from easily available starting materials. Mechanistic studies revealed the process was initiated by single electron oxidation of alkenes, followed by an intermolecular nucleophilic attack and intermolecular radical trap to form a C−C bond, representing an intermolecular trap of this carbon radical resulting from radical cation of alkenes to form C−C bonds. We anticipate this reaction will pave the way for this reaction mode to discover new reactions and open the avenue for catalytic transforming inorganic ammonium salts into N-containing organic frameworks.

## Methods

### General procedure A for the photocatalytic selective [1 + 2 + 3] construction of 2-piperidinones from alkenes and ammonium salt

Under an inert atmosphere, an oven-dried Schlenk-tube equipped with a magnetic stir bar was charged with N-Ph-9-mesityl 3,6-di-*tert*-butylacridinium tetrafluoroborate (1.4 mg, 2.5 μmol, 2.5 mol%), NH$_4$OAc (23.1 mg, 0.3 mmol), LiBF$_4$ (9.4 mg, 0.1 mmol) and alkene (if solid, 0.1 mmol), CH$_3$CN (1.0 mL), alkene (if liquid, 0.1 mmol), acceptor (0.2 mmol) and PhCl (0.1 mL) were added consecutively via syringe. The tube was sealed with a Teflon-coated septum cap, and stirred at ambient temperature under irradiation with 30 W blue LEDs for 12 or 24 h. Upon completion, the reaction mixture was quenched with water and extracted with ethyl acetate. The combined organic phase was concentrated in vacuum. The crude mixture was analyzed by $^1$H NMR with PhTMS as internal standard to determine the conversion and was directly purified by column chromatography on silica gel to give the corresponding compound.

### General procedure B for the photocatalytic selective [1 + 2 + 3] construction of 2-piperidinones from alkenes and ammonium salt

Under an inert atmosphere, an oven-dried Schlenk-tube equipped with a magnetic stir bar was charged with N-Ph-9-mesityl 3,6-di-*tert*-butyla-cridinium tetrafluoroborate (2.9 mg, 5 μmol, 5 mol%), NH$_4$OAc (23.1 mg, 0.3 mmol), LiBF$_4$ (9.4 mg, 0.1 mmol) and alkene (if solid, 0.1 mmol), CH$_3$CN (1.0 mL), alkene (if liquid, 0.1 mmol), acceptor (0.2 mmol) and PhCl (0.1 mL) were added consecutively via syringe. The tube was sealed with a Teflon-coated septum cap, and stirred at ambient temperature under irradiation with 30 W blue LEDs for 24 h. Upon completion, the reaction mixture was quenched with water and extracted with ethyl acetate. The combined organic phase was concentrated in vacuum. The crude mixture was analyzed by $^1$H NMR with PhTMS as internal standard to determine the conversion and was directly purified by column chromatography on silica gel to give the corresponding compound.

**General procedure C for the photocatalytic selective [1 + 2 + 3] construction of 2-piperidinones from alkenes and ammonium salt**

Under an inert atmosphere, an oven-dried Schlenk-tube equipped with a magnetic stir bar was charged with *N*-Ph-9-mesityl 3,6-di-*tert*-butyla-cridinium tetrafluoroborate (2.9 mg, 5 μmol, 5 mol%), $NH_4OAc$ (23.1 mg, 0.3 mmol), $LiBF_4$ (9.4 mg, 0.1 mmol) and alkene (if solid, 0.1 mmol), $CH_3CN$ (10.0 mL), alkene (if liquid, 0.1 mmol), acceptor (0.2 mmol) and PhCl (0.1 mL) were added consecutively via syringe. The tube was sealed with a Teflon-coated septum cap, and stirred at ambient temperature under irradiation with 30 W blue LEDs for 24 h. Upon completion, the reaction mixture was quenched with water and extracted with ethyl acetate. The combined organic phase was concentrated in vacuum. The crude mixture was analyzed by $^1H$ NMR with PhTMS as internal standard to determine the conversion and was directly purified by column chromatography on silica gel to give the corresponding compound.

## Data availability

The experimental data and the characterization data for all the compounds generated in this study have been provided in the Supplementary Information. Crystallographic data for the structures reported in this paper have been deposited at the Cambridge Crystallographic Data Centre, under deposition numbers 2194532 (**3a**), 2194539 (**3y**), 2194535 (**4j**), 2207224 (**4o**), 2194544 (**5j**), 2194546 (**5o**), 2201531 (**6a**), 2194547 (**6b**), 2194548 (**6e**) and 2194553 (**7c**). These data can be obtained free of charge from The Cambridge Crystallographic Data Centre via www.ccdc.cam.ac.uk/data_request/cif.

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

## Acknowledgements

W.S. gratefully acknowledges the financial support from NSFC (22371115, 21971101, 22171127), Guangdong Basic and Applied Basic Research Foundation (2022A1515011806), Department of Education of Guangdong Province (2022JGXM054, 2021KTSCX106), The Pearl River Talent Recruitment Program (2019QN01Y261), Shenzhen Science and Technology Innovation Committee (JCYJ20220519201425001), Thousand Talents Program for Young Scholars, Guangdong Provincial Key Laboratory of Catalysis (2020B121201002) is sincerely acknowledged. We thank Dr. Qiao Song (SUSTech) for the assistance of UV-vis spectrophotometer analysis. We acknowledge the assistance of SUSTech Core Research Facilities. This work is dedicated to the 100th birthday of Prof. Li-Xin Dai.

## Author contributions

W.S. conceived and directed the project. Y.D.D. discovered and developed the reaction. Y.D.D., S.W., and X.Y.C. (X-Yi.C) performed the experiments and collected the data. H.W.D. and X.Y.C. (X-Yong.C.) performed the crystal structure determinations. Y.L.L. discussed the project with W.S.. W.S. wrote the paper with contributions from all authors. Y.D.D. and S.W. contributed equally.

## Competing interests

The authors declare no competing interests.
