## [Peer Review File · Nature Communications]

Organophotocatalysed [1+2+3] Strategy Enabled One-Step to 2-Piperidinones from Alkenes and Ammonium SaltEditorial Note: Parts of this peer review file have been redacted as indicated to maintain the confidentiality of other journals.

REVIEWERS' COMMENTS

Reviewer #1 (Remarks to the Author):

The authors have described a new photomediated three-component reaction involving an ammonium salt as an ammonia surrogate and two different alkene coupling partners for the single-step synthesis of substituted piperidinones. The method shows exclusive chemoselectivity regarding order of reaction of the two alkenes and generally results in moderate diastereoselectivities. A large substrate scope has been included showing a number of iterations upon the two alkene coupling partners. Substituted styrenes and various acrylates constitute the majority of the two alkene classes, but some extension to vinyl ethers and simple alkyl alkenes has been made. A few examples of more elaborate substrates derived from natural products have been included. Yields range from 31-99% with average yields being in the 50-70% range. Mechanistic studies have been included and a reasonable mechanism proposed.

I reviewed this manuscript previously for [reacted] and I find the current version much improved. All of my previous concerns have been adequately addressed with two exceptions (see below). I would especially praise the authors for their inclusion of more robust mechanistic analysis.

Two relatively minor issues that should be addressed in the current manuscript:

- a. The term “unprecedented” remains in the first sentence of the Discussion section. As mentioned in previous reviews, as well as in the authors’ responses, this term is not quite accurate and should be removed.
- b. In the Methods section, EA is still used for ethyl acetate but without any prior definition. An abbreviation should be defined before it is used.

After the above edits, I find the current manuscript of appropriate caliber for publication in Nature Communications.

Reviewer #2 (Remarks to the Author):

In this manuscript, Shu and co-workers describe an organophotocatalyzed [1+2+3] strategy for rapidly constructing substituted 2-piperidinone from easily available alkenes, which provides an unprecedented approach to access N-containing heterocycles. A wide range of 2-piperidinone derivatives was successfully obtained with satisfactory yields. This manuscript was reviewed five

months ago. According to comments, the authors supplied some experiments and carefully revised this manuscript. Thus, I recommend this manuscript published in Nature Communications.

Reviewer #3 (Remarks to the Author):

This referee is greatly appreciative of the comments given in answer to the original reviewers' comments.

The first referee commented, "there is no information (including citations) in the manuscript, or even in the ESI. With so many structures, all pictured (albeit small), I would expect some information in the manuscript." The Authors have responded to this with "descriptions of crystal structure are added in the revised manuscript." While this is an improvement, some of these comments are less than ideal. For example, a comment has been added to say, "the major isomers of compounds (4j, 4o, 5j, 5o, 6a, 6b, 6e, and 7c) were confirmed by X-ray diffraction analysis." Unfortunately, structures 4j, 5j & 5o, include disorder associated with inversion of the chirality (*vide infra*). These solid solutions may or may not be representative of the bulk - a different crystal could give the opposite hand or even a mixture of diastereomers. I would suggest these are rephrased to take that into account. Further, I would suggest some sort of very short General Procedure should be added along with the others in the Methods section, noting the presence of this disorder.

The disorder mentioned above has been treated extensively, but is not mentioned anywhere in either the manuscript or the ESI. Indeed, all the Figures show only the "desired component" of the structures which is unclear at best, and could be seen intentionally misleading the reader. The Figures need to specifically state where counterions, solvent or disorder have been omitted "for clarity".

The authors comment that "the "X-ray structure" have been replaced with "crystal structure" in the revised Supplementary Information - this should also be altered in the manuscript. This referee does not know what an X-ray looks like, but is quite confident that it doesn't look like any of the Figures in the manuscript!

The responses to Comments 4-6 & 11 give confidence that the crystal structure analysis has been done by people with knowledge, but this is information which should also be shared with readers. ESI is a fantastic opportunity to do so and this discussion and more should be included in both the Supplementary Information file, but also the `_refine_special_details` sections for each structure in

the CIF. Failure to do this makes it appear that the authors are hiding problems which are challenges associated with the samples. Comment 7 made it very clear that "a detailed discussion of the treatment of the

structures including any shortcomings should be included in the ESI and in the "refine_special_details" of each CIF as appropriate" - this has still not been done.

In the original version of the CIF, the `computing_data_collection` was missing - this has now been fixed, however, the CIF for 3a is now missing a substantial part of the meta-data. This includes a substantial part of the cell measurement data; the crystal size, shape & colour; estimated and experimental transmission factors; and the computing reduction and cell refinement information. Some meta-data for the other structures are also absent in particular experimental absorption details and transmission factors; the latter should be available from the data processing software.

While adding detail regarding the shortcomings of the data to the ESI, the authors might like to correct the spelling of ShelXT and ShelXL (they should be ShelXT and ShelXL - the Shel part is a reference to the author, George Sheldrick). Some of the formatting in tables S9 and S10 could also do with some work.

Reviewer #4 (Remarks to the Author):

In this revised version of the manuscript the authors have addressed many of the questions raised by the 4 reviewers in a satisfactory manner. Reviewer 2 and myself (Reviewer 3) pointed out that the lack of novelty and the restriction of the scope did not warrant publication in [redacted] however I do think that Nature Communication is more suitable for this revised version. The only comment that I have to this revision is that the mechanistic experiments do not rule out chain propagation in any way. The light on-off experiments and quantum yield are still consistent with chain propagation pathways actively generating product. However, the results do not support chain propagation. Consequently, the discussion should be phrased accordingly. I do not think that chain propagation is likely, and I do recommend that the authors rationalize this with a short discussion involving the redox potential of M3 and the substrates (Figure 5).

Overall, I think this to be a solid piece of work and I congratulate the authors for this valuable contribution to the field of method development.

Point-to-Point Response to Reviewers' Comments

Reviewer #1 (Remarks to the Author):

General comment: The authors have described a new photomediated three-component reaction involving an ammonium salt as an ammonia surrogate and two different alkene coupling partners for the single-step synthesis of substituted piperidinones. The method shows exclusive chemoselectivity regarding order of reaction of the two alkenes and generally results in moderate diastereoselectivities. A large substrate scope has been included showing a number of iterations upon the two alkene coupling partners. Substituted styrenes and various acrylates constitute the majority of the two alkene classes, but some extension to vinyl ethers and simple alkyl alkenes has been made. A few examples of more elaborate substrates derived from natural products have been included. Yields range from 31-99% with average yields being in the 50-70% range. Mechanistic studies have been included and a reasonable mechanism proposed. I reviewed this manuscript previously for [redacted] and I find the current version much improved. All of my previous concerns have been adequately addressed with two exceptions (see below). I would especially praise the authors for their inclusion of more robust mechanistic analysis.

General response: We thank the reviewer for supporting the publication of this work in *Nature Communications*!

Comment 1: Two relatively minor issues that should be addressed in the current manuscript: a. The term “unprecedented” remains in the first sentence of the Discussion section. As mentioned in previous reviews, as well as in the authors’ responses, this term is not quite accurate and should be removed.

Our response: As suggested, the term “unprecedented” in the first sentence of the Discussion section has been deleted.

Comment 2: b. In the Methods section, EA is still used for ethyl acetate but without any prior definition. An abbreviation should be defined before it is used. After the above edits, I find the current manuscript of appropriate caliber for publication in *Nature Communications*.

Our response: As suggested, the EA in the Methods section have been replaced with “ethyl acetate”.

Reviewer #2 (Remarks to the Author):

General comment: In this manuscript, Shu and co-workers describe an organophotocatalyzed [1+2+3] strategy for rapidly constructing substituted 2-piperidinone from easily available alkenes, which provides an unprecedented approach to access N-containing heterocycles. A wide range of 2-piperidinone derivatives was successfully obtained with satisfactory yields. This manuscript was reviewed five months ago. According to comments, the authors supplied some experiments and carefully revised this manuscript. Thus, I recommend this manuscript published in *Nature Communications*.

General response: We thank the reviewer for supporting the publication of this work in *Nature Communications*!

Reviewer #3 (Remarks to the Author):

Comment: This referee is greatly appreciative of the comments given in answer to the original reviewers' comments. The first referee commented, "there is no information (including citations) in the manuscript, or even in the ESI. With so many structures, all pictured (albeit small), I would expect some information in the manuscript." The Authors have responded to this with "descriptions of crystal structure are added in the revised manuscript." While this is an improvement, some of these comments are less than ideal. For example, a comment has been added to say, "the major isomers of compounds (4j, 4o, 5j, 5o, 6a, 6b, 6e, and 7c) were confirmed by X-ray diffraction analysis." Unfortunately, structures 4j, 5j & 5o, include disorder associated with inversion of the chirality (vide infra). These solid solutions may or may not be representative of the bulk - a different crystal could give the opposite hand or even a mixture of diastereomers. I would suggest these are rephrased to take that into account. Further, I would suggest some sort of very short General Procedure should be added along with the others in the Methods section, noting the presence of this disorder.

Our response: We thank the reviewer for critical comments!

In terms of compound **4j**, the disorder in chiral carbon atoms does indicate the presence of a certain proportion of isomers in the structure. However, the disorder does not affect the confirmation of relative configuration, as the disordered parts corresponds to two enantiomers, which can be confirmed by the ^1H and ^{13}C NMR analysis of **4j**.

For compounds **5j** and **5o**, one of the two chiral carbons involves disorder, resulting in two relative configurations in the structure. As result, the major isomer could not be assigned (which are also in consistent with the ^1H and ^{13}C NMR analysis). The crystal structures of **5j** and **5o** could only be used for confirming the structure.

Thus, we have deleted the crystal structure of **5j** and **5o** from Fig. 3 and the description in manuscript has been rephrased as "The major isomers of compounds (**4j**, **4o**, **6a**, **6b**, **6e**, and **7c**) were confirmed by X-ray diffraction analysis."

Finally, a detailed procedure has been added in Supplementary Information "For crystal structures of **4j**, **5j** and **5o**, the molecules in the asymmetric unit is disordered due to the presence of isomer in the sample. Two disordered parts have been solved by the "disorder tools" plugin in Olex2. All 1,2-distances and 1,3-distances involved in the disorder have been restricted by SADI command. In addition, EADP and/or ISOR commands have been used to make the anisotropic displacement parameters of disordered atoms reasonable. For more details, please see the CIF files.". For details, please see Supplementary Information Page 81.

Comment 2: The disorder mentioned above has been treated extensively, but is not mentioned anywhere in either the manuscript or the ESI. Indeed, all the Figures show only the "desired component" of the structures which is unclear at best, and could be seen intentionally misleading the reader. The Figures need to specifically state where counterions, solvent or disorder have been omitted "for clarity".

Our response: As suggested, we have added statements in Footnotes of Supplementary Figs. For details, please see:

Supplementary Fig.3, "The 2-Phenylacetamide molecule has been omitted for clarity."

Supplementary Fig.4, “The disordered part has been omitted for clarity.”

Supplementary Fig.5, “The disordered part and water molecules have been omitted for clarity.”

Supplementary Fig.6, “The disordered part has been omitted for clarity.”

Supplementary Fig.7, “The disordered part and water molecules have been omitted for clarity.”

Supplementary Fig.8, “The disordered part has been omitted for clarity.”

Supplementary Fig.12, “The disordered part and dichloromethane molecule have been omitted for clarity.”

Comment 3: The authors comment that "the “X-ray structure” have been replaced with “crystal structure” in the revised Supplementary Information - this should also be altered in the manuscript. This referee does not know what an X-ray looks like, but is quite confident that it doesn't look like any of the Figures in the manuscript!

Our response: As suggested, the “X-ray structure” has been replaced with “crystal structure” in the manuscript. For details, please see Figs. 2 and 3 in the revised manuscript.

Comment 4: The responses to Comments 4-6 & 11 give confidence that the crystal structure analysis has been done by people with knowledge, but this is information which should also be shared with readers. ESI is a fantastic opportunity to do so and this discussion and more should be included in both the Supplementary Information file, but also the `_refine_special_details` sections for each structure in the CIF. Failure to do this makes it appear that the authors are hiding problems which are challenges associated with the samples. Comment 7 made it very clear that "a detailed discussion of the treatment of the structures including any shortcomings should be included in the ESI and in the "refine_special_details" of each CIF as appropriate" - this has still not been done.

Our response: As suggested, the detailed discussions have been added in the ESI and CIF files.

1. For details in the ESI file, please see Page 81 “The crystal structure of **3y** is slightly disordered due to atom vibration, the configuration determination of **3y** is not affected by the disorder. For crystal structures of **4j**, **5j** and **5o**, the molecules in the asymmetric unit is disordered due to the presence of isomer in the sample. Two disordered parts have been solved by the "disorder tools" plugin in Olex2. All 1,2-distances and 1,3-distances involved in the disorder have been restricted by SADI command. In addition, EADP and/or ISOR commands have been used to make the anisotropic displacement parameters of disordered atoms reasonable. For more details, please see the CIF files.”

2. The detailed discussions have also been added in the revised cif files.

Take **3y** as an example, the details of the CIF modification are shown below:

```
_cell_measurement_theta_max    27.52
_cell_measurement_theta_min    2.89
_shelx_estimated_absorpt_T_max 0.981
_shelx_estimated_absorpt_T_min 0.975
_exptl_absorpt_coefficient_mu  0.079
_exptl_absorpt_correction_T_max 0.745
_exptl_absorpt_correction_T_min 0.450
_exptl_absorpt_correction_type multi-scan
_exptl_absorpt_process_details
:
```

_reflns_special_details;

Reflections were merged by SHELXL according to the crystal class for the calculation of statistics and refinement.

_reflns_Friedel_fraction is defined as the number of unique Friedel pairs measured divided by the number that would be possible theoretically, ignoring centric projections and systematic absences.

The structure is slightly disordered, and the disorder has been described by formulating two different positions per disordered atom. EADP, SADI and ISOR constraints/restraints have been used to keep the refinement stable and reasonable.

For more detail, please see updated CIF files for **3a**, **4j**, **5j**, **5o**.

Comment 5: In the original version of the CIF, the computing_data_collection was missing - this has now been fixed, however, the CIF for 3a is now missing a substantial part of the meta-data. This includes a substantial part of the cell measurement data; the crystal size, shape & colour; estimated and experimental transmission factors; and the computing reduction and cell refinement information. Some meta-data for the other structures are also absent in particular experimental absorption details and transmission factors; the latter should be available from the data processing software.

Our response: As suggested, the computing data_collection for **3a** has been added, see updated CIF file for detail.

Comment 6: While adding detail regarding the shortcomings of the data to the ESI, the authors might like to correct the spelling of SheIXT and SheIXL (they should be ShelXT and ShelXL - the Shel part is a reference to the author, George Sheldrick). Some of the formatting in tables S9 and S10 could also do with some work.

Our response: We thank the reviewer for pointing out this issue! The SheIXT and SheIXL in supplementary information has been replaced with ShelXT and ShelXL. In addition, the data in Supplementary Tables 9 and 10 (previous Tables S9 and S10) have been reformatted as suggested.

Reviewer #4 (Remarks to the Author):

Comment: In this revised version of the manuscript the authors have addressed many of the questions raised by the 4 reviewers in a satisfactory manner. Reviewer 2 and myself (Reviewer 3) pointed out that the lack of novelty and the restriction of the scope did not warrant publication in [redacted] however I do think that Nature Communication is more suitable for this revised version. The only comment that I have

to this revision is that the mechanistic experiments do not rule out chain propagation in any way. The light on-off experiments and quantum yield are still consistent with chain propagation pathways actively generating product. However, the results do not support chain propagation. Consequently, the discussion should be phrased accordingly. I do not think that chain propagation is likely, and I do recommend that the authors rationalize this with a short discussion involving the redox potential of M3 and the substrates (Figure 5). Overall, I think this to be a solid piece of work and I congratulate the authors for this valuable contribution to the field of method development.

Our response: We thank the reviewer for supporting the publication of this work in *Nature Communications*! The highly oxidizing acridinium photoredox catalyst Mes-3,6-*t*Bu₂-Acr-Ph⁺BF₄⁻ ($E^*_{\text{red}} = +2.10$ V vs SCE, $E_{1/2}(\text{Acr}^+/\text{Acr}^\cdot) = -0.57$ V vs SCE.) could oxidize alkenes to generate cation radicals **M1**. For terminal alkenes attached with a phenyl ring, the oxidation potential is below 2.0 V, such as 1.97 V for styrene and 1.91 V for α -methyl styrene. For aliphatic alkenes, the oxidation potential is 1.98 V for 2-methylbut-2-ene, 1.75 V for 1-methylcyclopentene, 1.51 V for 3,4-dihydro-2H-pyran (*Synlett* **2016**, 27, 714 – 723). Unfortunately, we do not get the exact redox potential of the radical intermediate **M3**. However, the similar alkyl radical intermediate generated from alkyl radical addition to acrylate which can oxidize the reduced form of the acridinium catalyst **PC-1** to regenerate **PC** (*J. Am. Chem. Soc.* **2018**, 140, 4213 – 4217; *J. Am. Chem. Soc.* **2018**, 140, 9056–9060; *ACS Catal.* **2022**, 12, 2045 – 2051). Accordingly, the discussion is rephased as “The quantum yield (Φ) of the reaction using **1b** and **2b** was determined to be 0.75 (Fig. 4e), indicating the reaction may undergo a catalytic radical process. Yet, a slow chain propagation mechanism cannot be ruled out at this stage.” in the revised manuscript.

We thank all the reviewers for insightful suggestions and comments!